# DICER1: The Argonaute Endonuclease Family Member and Its Role in Pediatric and Youth Pathology

**DOI:** 10.3390/biology14010093

**Published:** 2025-01-18

**Authors:** Consolato M. Sergi, Fabrizio Minervini

**Affiliations:** 1Division of Anatomic Pathology, Children’s Hospital of Eastern Ontario, Ottawa, ON K1H 8L1, Canada; 2Department of Pathology and Laboratory Medicine, University of Ottawa, Ottawa, ON K1N 6N5, Canada; 3Department of Laboratory Medicine and Pathology, University of Alberta, Edmonton, AB T6G 2B7, Canada; 4Division of Thoracic Surgery, Cantonal Hospital Lucerne, 6000 Lucerne, Switzerland; fabriziominervini@hotmail.com

**Keywords:** DICER1, oncogene, biology, molecular genetics

## Abstract

In 2001, two genes that code for enzymes were found in the fruit fly. These genes make enzymes that eat slightly different types of substrates. DICER1, the gene that is found in humans, is very important for healthy human growth. This enzyme works in a number of ways, such as blocking RNA, fixing DNA damage, and fighting viruses. DICER is also involved at the protein level in several human diseases. The pleuro-pulmonary blastoma is likely the most serious of these. A lot of research has been conducted to find out all of DICER1’s functions and how they relate to both cancer-causing and non-cancerous illnesses. This enzyme is found all over the body, which makes it a great target for nanotechnology-based treatments and drug repurposing.

## 1. Introduction

Despite significant interest and research, DICER1 is a relatively new and developing factor in human disease. This review gives a thorough overview of the gene, encompassing its protein structure, associations, and involvement in disease. The prospect of targeting DICER1 and endoribonuclease DICER as therapeutic interventions is highly appealing. Through additional research, we may effectively utilize their widespread presence to treat many diseases. The *DICER1* gene encodes a crucial protein in various biological processes. It is involved in producing small RNA molecules, which are essential for regulating gene expression and controlling cellular functions. Two genes encoding enzymes were discovered in *D. melanogaster* in 2001 [1]. The genes *Dicer-1* and *Dicer-2* produce ribonuclease enzymes. They target quite different substrates. The human orthologue is *DICER1*. It is a gene found on chromosome 14q32.13. *DICER1* consists of 27 exons. It possesses two domains [2]. This gene exhibits expression in tissues derived from several organ systems and proper human development relies on this key gene. The protein encoded by *DICER1* is an endoribonuclease. It belongs to the RNase III superfamily, also known as DICER. This set of proteins is found ubiquitously in eukaryotes. Prokaryotes lack an analogous protein, but they possess proteins that operate in comparable paths [3]. In general, the primary role of RNase III proteins is to identify and split apart double-stranded molecules of RNA [3].

In humans, the DICER molecule has an “L” shape consisting of a head, body, and base. The substrate, often dsRNA (double strand RNA) or pre-miRNA, is located in the main body of the molecule (Figure 1).

A helicase domain, two RNase III domains (RNase IIIa and RNase IIIb), a double-stranded RNA binding domain, a PAZ (Piwi–Argonaute–Zwille) domain, and a domain of an undetermined function (DUF283) are among the recognized functional domains [5,6,9]. Characterizing DICER and its domains in detail has proven difficult due to the protein’s considerable size. Most domains’ major activities and structural placements have been clarified by utilizing diverse study methods and homologs in other organisms (Figure 2 and Figure 3).

The molecule contains two critical domains, such as the RNase IIIa and IIIb. The domains are the operational components responsible for cleaving dsRNA molecules. The helicase domain, which is located at the base of the protein, works in conjunction with the dsRNA binding domain (dsRBD) to identify and attach to the substrate. The domain, labeled as PAZ, is located at the N-terminus of the protein. It identifies the 5’ phosphate and two specific nucleotides of the substrate molecules. It assists in their precise positioning. The function of DUF283 is currently ambiguous, although it has been proposed to play a task in the binding of RNA to other (protein) complexes and in enhancing the effectiveness of enzyme action [10,11,12].

The cleavage effect mediated by *DICER* produces two main products. They are micro-RNA (miRNA) and small interfering RNA (siRNA). MiRNA maturation commences in the nucleus by synthesizing a lengthy primary transcript RNA molecule (pri-miRNA). The pri-miRNA molecule is subsequently cleaved by an enzyme complex containing DROSHA, which is a member of the superfamily of RNase III. This cleavage results in a pre-miRNA molecule nearly 60 bases in length and possesses a hairpin structure. Exportin 5 facilitates the carriage of this molecule toward the cytoplasm from the nucleus, where it can be subsequently identified and cleaved by *DICER* to generate miRNA. On the other hand, siRNA molecules are generated through the process of DICER cleaving various double-stranded RNA precursors. The enzyme’s structure and its domains, which have been labeled as the ’molecular ruler’, ensure that both miRNA and siRNA molecules are constantly 20–24 bases long.

Ago and the transactivating response RNA-binding protein (TRPB) are two more proteins that accompany the enzyme-substrate complex after DICER cleavage. The Argonaute (AGO) gene family provides instructions for making six distinct protein domains: N-terminal (N), Linker-1 (L1), PAZ, Mid, Linker-2 (L2), and a PIWI domain at the very end. The entire DICER–Ago–TRBP complex is commonly known as the ’RISC-loading complex’, responsible for loading the short double-stranded RNA products into another protein complex known as the RISC (RNA-induced silencing complex). The resulting combination of RISC and dsRNA is called miRISC or siRISC, depending on the specific type of tiny dsRNA present. This mechanism marks the initiation of the RNA interference (RNAi) pathway [13,14].

The physiological function of DICER in humans is complex. DICER1 is ubiquitously expressed and is an essential housekeeping gene in physiological tissue development. Eight short RNA products produced by DICER play a role in several activities. Therefore, *DICER* is likely involved in more cellular functions than we know. DICER plays a function in miRNA processing and participates in DNA processing, remodeling of chromatin granule structure, and programmed cell death or apoptotic pathways (Table 1).

## 2. RNA Interference

Post-transcriptional gene silencing, also known as RNA interference, is a mechanism that regulates the expression of genes that code for proteins. After forming the RISC-loading complex, it binds to the RISC, which is essentially the executor of the RNA interference paths. Upon activation of this complex, dsRNA undergoes separation. The passenger strand is eliminated and undergoes degradation, while the other strand persists in the RISC and functions as the guide strand. The RISC may now specifically attach to target RNA sequences, such as mRNA, and exert control over their translation or facilitate their breakdown. If the guide and the target have a perfect match, typically with siRNA, the Ago molecules (proteins) in the complex will utilize their endonuclease function to degrade the target, inhibiting translation. In the case of a flawed match, particularly with miRNA, the mRNA stays stable, but the translation process is inhibited. Perfect matching generally enables the selection of more precise objectives, whereas imperfect matching allows for a broader range of targets. The role of nuclear activities in repairing DNA harm is still under intense investigation. Contrary to earlier beliefs, new research has demonstrated that human DICER, thought to be only found in the cytoplasm, can be moved to the nucleus in reaction to specific stimuli. Typically, when DNA breaks into two strands, it activates the DICER protein through the ERK pathway, a member of the KRAS/MAPK system. This alteration of the protein facilitates translocation to the nucleus [10,11,12,15]. Upon arrival, DICER undergoes phosphorylation, generating short RNA molecules from the regions neighboring the chromosomal breakage site. The precise mechanism beyond this juncture remains incompletely comprehended. However, investigations conducted on cells lacking The DICER1 study found that the DNA damage repair mechanism is less effective than before. The accumulation of DNA breaks in both strands and an increased susceptibility to radiation from ultraviolet and gamma rays are both outcomes of this [16]. There are a number of other possible roles for this molecule in the nucleus, including repairing nucleotide excision, preserving heterochromatin structure, timing DNA replication, suppressing transposable elements, and facilitating apoptosis and autophagy [17].

## 3. Antiviral Properties

DICER primarily acts on endogenous dsRNA but also plays a crucial part in antiviral pathways across several taxa, comprising invertebrates, vertebrates, and plants. It is established that, in humans, it exhibits antiviral properties against some RNA viruses [10,11,12,13,14,15,16,17,18,19,20,21,22]. DICER can convert RNA of a virus into small RNA molecules called virus-derived small RNA molecular structures, or vsiRNAs, just like it does with endogenous dsRNA during normal physiological activity. Subsequently, these diminutive molecules can be employed comparably to miRNAs, wherein they bind and repress viral genes imperative for replication. Although the precise mechanism remains incompletely understood, research has demonstrated that DICER experiences downregulation in monocytes/macrophages during the infection of Human Immunodeficiency Virus 1 (HIV-1). Moreover, inhibiting DICER can enhance Adenovirus multiplication, further confirming this hypothesis [17]. DICER is also linked to the immune response against specific viruses, including Zika virus (a member of the virus family Flaviviridae primarily spread by mosquitoes) and SARS-CoV-2 (the etiologic agent for COVID-19 pandemics) [23,24,25]. There is a possibility that this system could be controlled by other viruses and ultimately generate pro-viral effects [26].

## 4. DICER1 in Human Disease and Its Pathobiology

The primary pathological function of DICER1 appears to be associated with the deregulation of miRNAs and, of course, the RNAi mechanism. *DICER1* gene mutations can be classified as either germline or somatic. Germline mutations can arise at any location within the gene, often leading to loss of function. On the other hand, somatic mutations tend to occur in specific ’hotspots’ [27,28,29,30,31]. As our understanding of the complete range of DICER1 function improves, we may uncover more nuanced roles in disease processes and, ideally, better comprehend the diseases themselves. Table 2 displays a compilation of specific disorders with which DICER1 has been associated. There is a concise overview of many disorders where dysregulation (abnormal regulation) or molecular modification of DICER1 have been linked, along with their postulated causes and specific discoveries related to the disorders. DICER1 has been demonstrated to participate in a wide range of pathological diseases. Occasionally, DICER appears to operate correctly, yet it receives an atypical substrate. Trinucleotide repeat diseases include the formation of hairpin-like structures due to the presence of elongated chains of repeats. These complexes, like the standard substrate of the DICER molecule, undergo processing and aggregate in cells, resulting in cellular damage [21,32]. Alternatively, the disease state is triggered in certain instances by the molecule’s intracellular amounts, as observed in a specific kind of age-linked macular degeneration known as “geographic atrophy”. The absence of DICER in the retinal pigmented epithelium (RPE) is critical, because individuals with geographic atrophy may show the buildup of a harmful protein transcript, ultimately causing vision loss [33]. In fact, Saeki et al. [34] examined DICER and its co-RNase, DROSHA. They proposed that there may be a connection between particular genetic variations and autoimmune thyroid diseases, such as Graves’ disease and Hashimoto thyroiditis, and that the involvement of the NLRP3 inflammasome is still puzzling in a similar situation, as seen in other diseases [35,36,37,38,39,40]. Their investigation also demonstrated a potential correlation between the levels of expression of these two proteins and the refractile/remission rate.

Furthermore, a longstanding correlation exists between thyroid disease, the multinodular goiter (MNG), and *DICER1* mutations. Below, the section on DICER1-related illnesses will contain further details. While the common assumption is that pediatric goiters are associated with *DICER1* gene mutations, individuals with these mutations can be diagnosed with goiters in their forties or later, particularly if they have not suffered from any other DICER1-related disorders. DICER has been identified as a potential cause in various diseases, such as neuropsychiatric disorders, cardiovascular disease, and fertility problems [17]. The limited occurrences presented demonstrate the diverse ways DICER1 may influence or trigger disease, and much study is now underway.

Neoplasia refers to the abnormal growth and proliferation of cells, leading to tumor formation. A major obstacle in comprehending the role of DICER changes and dysregulation in human disease lies in the incomplete understanding of the subsequent pathways involved. Although numerous diseases have exhibited DICER-related alterations, there is a lack of consistent patterns, particularly concerning neoplasia [26]. As mentioned, DICER’s impact on disease is mainly linked to the disruption of miRNAs, heading to alterations in overall genetic expression. Typically, cancerous growths have a decrease in the activity of small RNA molecules called miRNAs, and the absence of a properly functioning protein called DICER impairs the ability of cells to repair DNA damage [16]. DICER1 has been linked to the promotion of both benign and malignant neoplastic processes (tumors), and a hyperplastic and cystic phenotype characterizes the usual histological presentation, although this is not universally observed.

It has been suggested that *DICER1* is a haplo-insufficient gene that suppresses tumor growth [41]. In haplo-insufficiency, one gene copy is either deleted or rendered inactive, leaving just one functioning copy that is inadequate to produce the required gene product and maintain normal function. However, its precise categorization remains a subject of ongoing discussion and remains controversial [26,42]. The carcinogenesis associated with DICER1 has primarily been investigated in people with pathogenic germline gene mutations, resulting in the total loss of one allele of *DICER1* (refer to the section on DICER1 syndrome). Abnormal characteristics occur because of a second (somatic) mutation. What is remarkable, though, is that the secondary mutation does not lead to the elimination of the second *DICER1* copy (complete elimination of DICER1 will hinder cell growth and lead to cell death). Tumorigenesis requires a second mutation, typically a missense genetic mutation, which hampers the functionality of the DICER enzyme. Normally, the second impact strikes inside one of the five hotspot regions in the RNase IIIb region of the gene, resulting in a modification of the gene’s capacity to break down substrate molecules. The outcome is an atypical combination of miRNAs within the cell, leading to alterations in gene expression [41]. The variation in the composition of miRNAs leads to variation in the resulting illness phenotype.

Although most DICER1-related thyroid lesions are non-cancerous, there have been observations of their presence in thyroid carcinomas, which can vary in their level of differentiation from well differentiated toward a very aggressive phenotype. Germline mutations in *DICER1* that occur in both alleles are linked to a significantly higher risk of thyroid cancer, roughly 16 times more than the average risk. However, the risk seems modest in patients harboring a heterozygous germline genetic mutation. Oncogenesis is promoted in heterozygous mutations when there is a second promoting mutation in *DICER1* or another gene. Similar genetic alterations have been observed in other well-known thyroid oncogenes, for instance *BRAF* and *RAS*.

Importantly, Canbrek et al. conducted a review and found that about one individual among fifty people (1.5–3.7%) harboring thyroid carcinomas exhibit a *DICER1* mutation. However, the relevance of many of these mutations remains uncertain [43]. DICER1 has been found to share a regulatory transcription factor with TERT in vitro, based on investigations conducted on thyroid cancer. These investigations demonstrated a correlation between reduced amounts of the transcription factor GABPA (GA repeat binding protein alpha) and lower levels of DICER, as well as an increase in tumor invasion, neoplastic cell proliferation, and overall tumor call viability. The precise mechanisms responsible for these discoveries remain uncertain [41,42,44].

DICER seems also to have a crucial role in sporadic cancer instances. In events when heredity is not a factor, somatic genetic mutations arise in the identical hotspot region of the RNase IIIb gene, leading to modified protein functionality [45]. The initial *DICER1* genetic mutation can then be linked to a secondary genetic mutation, which can be either another somatic genetic mutation, a “mosaic landscape”, or a distinct germline genetic mutation to facilitate tumor development [43]. Phosphorylation of Dicer in certain animal studies was found to be linked with elevated levels of tissue and lympho-vascular invasion. When combined with other genetic mutations, for example in *KRAS* and *p53*, mice exhibited a more comprehensive range of neoplastic processes, which were also more likely to spread to numerous organs. Several investigations involving the suppression of *DICER1* gene expression demonstrated tumors developing heightened invasiveness and faster growth rates [15].

## 5. Disorders Associated with DICER1 Gene Mutations

Before the discovery and detailed classification of RNA interference and DICER, researchers were investigating cases of autosomal dominant (AD) inherited (familial) MNG. These family units had an abnormally high prevalence of individuals with strumae (goiters) that typically occurred throughout childhood or youth (early adulthood). It is now understood that benign thyroid lesions are the most prevalent indication of inherited *DICER1* gene mutations [27,28,46,47,48]. Delahunt and colleagues documented the initial familial correlation between pleuropulmonary blastoma and cystic nephroma in 1993 [49]. Subsequently, a greater number of tumors, both non-cancerous and cancerous, have been linked to inherited DICER1 defects, including disorders characterized by mosaicism, with varying degrees of manifestation.

Similar to other hereditary neoplasia-related conditions, individuals with this ailment are typically diagnosed with lesions at a younger age and usually require ongoing monitoring. For individuals with a specific set of genetic mutations known as “mosaicism for RNase IIIb domain hotspot mutations”, rigorous surveillance protocols are advised. This group typically exhibits a higher degree of genetic penetrance, with the manifestation of the disease occurring at a younger age and a broader range of observable characteristics. Currently, there have been no known specific associations between changes in the *DICER1* gene and observable characteristics [50].

## 6. DICER1 Syndrome

DICER1 syndrome, which stands for DICER1-related pleuropulmonary blastoma cancer predisposition syndrome, is passed down through families in a reduced-penetrance autosomal dominant fashion (Figure 4 and Figure 5). Although individuals with this genetic syndrome have an elevated risk of acquiring tumors, particularly thyroid carcinoma, most patients can expect to lead healthy lives. The frequency of germline loss of function (LOF) mutations in the *DICER1* gene is roughly 1 in 10,600, with merely a tiny fraction of these newly occurring [51]. Individuals affected by DICER1 syndrome experience the growth of both non-cancerous and cancerous tumors, with the majority of these tumors appearing before reaching the age of 40.

Most often, germline mutations manifest as “nonsense, frameshift, or splice site” genetic mutations, as well as minor deletions and insertions. These mutations might potentially occur at any location within the gene. As mentioned in the previous segment on neoplasia, it has been proposed that the development of tumors in DICER1 syndrome is caused by the inactivation of one allele of the *DICER1* gene in the germline, followed by the acquisition of a somatic mutation in the lasting allele. This mutation modifies the exons that contain the RNase IIIa and IIIb domains, leading to a decrease in the production of miRNAs. The thyroid, lungs, kidneys, and ovaries are the primary organs most impacted by DICER1 syndrome. The most prevalent manifestation, in general, is MNG. By the age of 20, 32% of females and 13% of males with DICER1 syndrome will have acquired MNG. These percentages rise to 40% and 17%, respectively, by age 40 years. Pleuropulmonary blastoma is commonly regarded as the most frequent cancerous occurrence, occurring in approximately 30–40% of individuals with DICER1 disease. This neoplasm primarily affects young children and is usually diagnosed before the age of six years. It can either manifest as benign cystic lesions (type Ir) or as an aggressive malignancy of mesenchymal type (sarcomatoid neoplasm) (type III). The prevalent diagnoses in the ovaries and kidneys are Sertoli-Leydig cell tumor of the ovaries and cystic nephroma of the kidney, respectively, among several other neoplasms [52,53,54,55,56].

Additional non-tumoral abnormalities have also been documented in individuals with DICER1 syndrome. The mentioned characteristics encompass macrocephaly, visual abnormalities, kidney and renal collecting system structural anomalies, and odontodysplasia or dental deformities [52,53]. Although specific abnormalities associated with DICER1 syndrome, such as thyroid lesions, are relatively common in the general community, people with this syndrome tend to develop these diseases at an earlier age than the general population. Pediatric populations have been observed to exhibit poorly differentiated thyroid cancer, which is usually found in older individuals and is related to DICER1 mutations [54]. Most of the remaining malignancies associated with DICER1 syndrome are uncommon, which theoretically facilitates the discovery of patients and families for screening purposes. Schultz et al. present a comprehensive overview of these tumors, along with recommendations for determining the diagnosis of DICER1 syndrome [55]. Other rare tumors have been described in individuals harboring DICER1 syndrome [56]. They include cranial (central nervous system) and extracranial (extra-central nervous system) neoplasms. Cranial neoplasms include medulloblastoma, ciliary body medulloepithelioma, pineoblastoma, and pituitary blastoma. Extra-cranial neoplasms include embryonal rhabdomyosarcoma, botryoid rhabdomyosarcoma of the cervix, gynandroblastoma of the ovary, gonadal granulosa cell tumor, gonadal choriocarcinoma, differentiated thyroid carcinoma, nasal chondromesenchymal hamartoma, exocrine and endocrine pancreatic carcinoma, ovarian fibrosarcoma, and renal sarcoma and peritoneal sarcoma. All these neoplastic processes arise from a germline mutation in the *DICER1* gene, leading to a predisposition for developing these tumors in different organs like the lungs, kidneys, ovaries, thyroid, pancreas, and the central nervous system. Their pathology does not differ from tumors originating in individuals without *DICER1* gene mutation.

## 7. GLOW Syndrome

The term GLOW syndrome stands for ***G***lobal developmental delay (psychological/psychometric delay), ***L***ung cysts, ***O***vergrowth (somatic), and ***W***ilms tumor or nephroblastoma, a disontogenetic neoplasm, which has been associated with nephrogenic rests [57] (Figure 6). Most recently, Wilms tumor has been associated with an exposure to parental contact to pesticides, especially in household settings and with organophosphate compounds [58].

GLOW syndrome was initially introduced by Klein et al. in 2014, and then further explored [42,59,60,61,62]. The authors showcased two instances of children exhibiting global developmental delay (psychologic/psychometric) and somatic overgrowth disorders in which mosaic DICER1 expression was seen. The tissue expression level varied between 21% and 47%, provisional on the patient and the kind of tissue. Subsequently, both pediatric patients experienced the development of pulmonary cysts and Wilms tumors, exhibiting a phenotype comparable to other disorders characterized by excessive tissue growth and a vulnerability to malignancies. Lung cysts should be a red flag, indicating that the pediatrician should think about abnormal development with or without DICER1 involvement at the first possibility he or she meet the patient. Nephroblastoma may usually occur in the 2nd infancy or later childhood [56].

In these cases, genetic research revealed a mosaic pattern of a newly occurring DICER1 mutation. This mutation was found in specific locations inside the RNase IIIb domain, which is recognized to be associated with “metal binding”. In both cases, the Wilms tumors and intact kidney tissue had a greater mutation prevalence than blood cells. Upon examination of the Wilms tumor in one patient, it was found that additional somatic mutations were explicitly present in the tumor tissue. However, the same mutations were not observed in the other case. Overall, GLOW syndrome should be considered as one of the DICER1-related illnesses. They postulate that specific genetic mutations in DICER1, particularly those that affect “metal-binding” sites in the Rnase IIIb domain, may enable DICER1 to act as an oncogene. No definitive mechanism exists to elucidate the precise locations implicated (or not implicated) in GLOW syndrome [42,59,60,61,62].

## 8. Assessing for *DICER1* Gene Mutations: Timing and Methodology

Although DICER expression appears widespread, the reasons for the differential impact on various organs and tissues remain unknown. There is no evident connection between the various cancerous growths other than a casual preference for a less developed or embryonic differentiation. However, it is essential to mention that many non-neoplastic disorders linked to alterations in DICER expression are recognized to exhibit a wide range of anatomical indications and symptoms. Specific lesions exhibit a strong correlation with *DICER1* gene mutations, thereby necessitating testing. Furthermore, it is crucial to assess when pediatric patients exhibit typical thyroid lesions commonly seen in adults, such as MNG or poorly differentiated thyroid cancer [28,46,47,63,64]. Germline *DICER1* mutations predominantly manifest as deletions or duplications/insertions, with occasional occurrences of translocations, albeit infrequently [17]. Copy number variations are the most frequent genetic changes observed in sporadic malignancies. These modifications often lead to increased expression of the DICER1 protein.

Additionally, missense mutations are very commonly found. Given the very limited occurrence of malignant tumors, identifying a disease-causing mutation in the body’s cells should at least lead to suspicion of a mutation in the reproductive cells [65].

When assessing an individual for DICER1 syndrome, molecular genetic testing may be specifically targeted depending on the patient’s profile. If the symptoms match those typically seen in DICER1 syndrome, conducting single testing or employing a targeted genetic panel may be beneficial. If the phenotype is less clear-cut, genetic research may be required.

As a general guideline, it may be critical to initiate the search by focusing on missense and nonsense genetic mutations, as well as insertions/deletions and splice site variants. If no anomaly is detected, it may be essential to run tests for more extensive deletions or duplications. Utilizing an inherited cancer multigene-based panel is an effective strategy to optimize the accurate detection of cancer genes and minimize expenses and potential complications, such as ambiguous genetic variations. Around one in 10 (10%) of patients with a new *DICER1* mutation will exhibit somatic *DICER1* mosaicism. In such instances, it may be necessary to conduct tests on various tissue types to definitively establish the existence of a harmful mutation. Furthermore, because DICER1-related sickness is very recent, it is typical to come across variations in unknown significance when evaluating these patients. De Kock, Wu, and Foulkes propose an advisable method for handling these circumstances [50].

## 9. DICER and Therapies

Currently, no specific therapies for cancers and illnesses with mutations in the *DICER1* gene have been established and consolidated. The gene’s involvement in numerous disease states makes it an appealing target. However, the challenge is finding its therapeutic potential without risking substantial side effects until the complete range of its functions is uncovered. Certain medications now in use have demonstrated the capacity to enhance DICER1 activity. Metformin has been shown to modify the distribution of DICER, resulting in an elevation of its cytoplasmic concentrations, as observed in both mice and human research. Enoxacin, a type of antibiotic called fluoroquinolone, has been found to enhance the activity of DICER [66]. In addition to antibiotic properties, enoxacin has also been demonstrated to successfully inhibit the growth of multiple neoplastic cells [67]. Fascinatingly, enoxacin has been conveyed to have a low incidence rate of toxicity due to its ability to selectively prevent the growth of neoplastic cells, while leaving non-neoplastic cells untouched. The enoxacin adjuvant therapy could potentially have a protective effect on the brain in Parkinson’s disease. The role of epigallocathechin gallate is particularly interesting and under intense investigation, particularly considering the protective effect of green tea against neurodegenerative diseases [68]. Interestingly, enoxacin and epigallocatechin gallate accomplish synergistically to inhibit the growth of cervical neoplastic cells [69]. Additionally, interferon-beta1 has been shown to raise the levels of DICER in the blood of certain people with multiple sclerosis who have had positive clinical outcomes [17]. The potential intersection with TERT pathways15 may also hold valuable therapeutic implications. However, further inquiry is required, as said earlier.

### 9.1. The Two-Hit Hypothesis: Insights into Molecular Mechanisms of DICER1 Mutations

The two-hit concept, which was formally postulated by Knudson in 1971 [70,71], was initially proposed in 1953. The theory proposes that abnormalities can only manifest when two distinct mutations occur in each allele, and that tumors cannot be caused by a single mutation in any one allele. The average person inherits a single mutation, which poses little health risk on its own. But it is also possible for two mutations to work together to cause cancer. Several instances of tumor suppressor gene deactivation can be explained by the two-hit concept [72,73]. It has been established that DICER1 syndrome is an inherited, haploinsufficient autosomal dominant illness [74,75]. Evidence from multiple DICER1 syndrome instances supports this hypothesized mechanism. These examples included DICER1 syndrome symptoms in people with a single apparent germline mutation. In contrast, new research shows that DICER1 syndrome individuals had a somatic mutation in the second *DICER1* gene allele in addition to inherited alterations in the first allele. When discussing *DICER1* mutations and Dicer’s function as a tumor suppressor gene, the two-hit hypothesis is relevant. The DICER1 gene’s RNase IIIb domain has second-hit somatic mutations. According to research including three Wilms’ tumor patients, the two-hit hypothesis was relevant to the development of Wilms’ tumor in patients with DICER1 syndrome [76]. Screening for somatic *DICER1* mutations revealed that the patients had germline mutations in the gene, and further testing revealed that the patients had somatic mutations in the RNase IIIb region on the second allele. This discovery also brings attention to the fact that Wilms tumor patients often have mutations in the RNase IIIb domain of the *DICER1* somatic gene. There are genetic hotspots for somatic mutations within the *DICER1* gene inside the areas encoding the RNase III domains [77,78]. The second mutation in pleuropulmonary blastomas often occurs in the RNase IIIb domain and is part of a biallelic *DICER1* mutation [79]. In research involving eleven patients with spontaneous pleuropulmonary blastomas and DICER1 gene mutations, eight of the patients had biallelic DICER1 gene mutations, with one mutation located in the RNase IIIb domain. An ovarian fibrosarcoma tumor from a 9-year-old girl was found to have biallelic *DICER1* mutations. The tumor also had a second point mutation, which caused a substitution at amino acid position 1813 within the RNase IIIb domain (p.E1813G) [80]. The tumor also had a germline single base insertion in the DICER1 gene, which caused a frameshift and premature stop codon.

In another instance, a female patient who was 14 months old at the time of diagnosis had pleuropulmonary blastoma in addition to a cystic nephroma that had been resected 11 months earlier [81]. The *DICER1* gene was found to be compromised in both alleles after mutation analysis was performed on peripheral blood and accessible tissue. The DNA sample from the cystic nephroma included a missense heterozygous somatic mutation (c.5425G>A; p.G1809R), and there was also a germline truncating mutation (c.5347C>T; p.Q1783*) in exon 24, which codes for the RNase IIIb domain, in the peripheral blood. The patient’s maternal lineage was found to carry this genetic mutation. A female cousin of the patient, who is 21 years old, was also discovered to carry the germline mutation. She underwent treatment for embryonal rhabdomyosarcoma when she was 14 years old and MNG when she was 20 years old. The patient’s embryonal rhabdomyosarcoma (c.5428G>C; p.D1810H) was found to have a novel missense heterozygous somatic mutation. This instance shows that the mechanism of DICER1 syndrome is not due to haploinsufficiency but rather to biallelic mutations in the *DICER1* alleles. Second, somatic, hot-spot alterations in the RNase IIIb domain were the subject of Brenneman and colleagues’ [82] discussion of biallelic *DICER1* gene mutations. The mutation status of DICER1 alleles was determined in a cohort of patients diagnosed with pleuropulmonary blastoma. It is possible that the rate-limiting phase in the pathophysiology of DICER1 syndrome involves mutations inside the RNase IIIb domain, which could constitute hotspot mutations [83]. Patients typically had DICER1 germline mutations that caused a loss of function, and while RNase IIIb hotspot mutations were uncommon outside of tumors, they were more common inside them. Because Dicer/DICER is involved in DNA replication and repair, patients with RNase IIIb mutations may also be at increased risk for other mutations.

### 9.2. Limitations

This review is comprehensive but has some limitations and we focused on the gross and microscopic examination of the pediatric tumors associated with DICER1 that we came across during our personal experience. On the other hand, several other neoplasms may be associated with DICER1 syndrome as described above. Probably, a current hot topic is the onset of pancreatic cancer associated with DICER1. We could not illustrate much, because the literature is scarce. Research on DICER1’s function in exocrine pancreatic cancer and endocrine pancreatic cancer is limited [84,85]. The process of acinar-to-ductal metaplasia (ADM) and epithelial-to-mesenchymal transition (EMT) in pancreatic acinar cells was shown to begin in mice with homozygous knockout of Dicer1 [86]. In 2014, the onset of pancreatic intraepithelial neoplasms (PanIN) was accelerated in a study by Wang et al. [86] and by Eser et al. [87], who found that oncogenic Kras, a small GTPase, promotes pancreatic neoplasia when activated simultaneously with heterozygous deletion of Dicer1. Kras activation in conjunction with Dicer1 homozygosity resulted in increased ADM formation but had no effect on PanIN initiation rate [86]. They also discovered that acinar cell regeneration was allowed in the absence of Kras activation after extensive Dicer1 deletion. Despite the importance of this discovery, it is not completely ruled out that other variables, like a population of progenitor cells or transdifferentiated cells, may have had a role in the pancreatic cell regeneration [86,88]. These results suggest that the exocrine pancreas tumor growth could be dose-dependent with regard to DICER1. To confirm the role of DICER1 in pancreatic cancer, additional research is needed to address potential technical limitations, such as the fact that *Dicer1*fl/fl experimental animals retain one functional *Dicer1* allele or the difficulty in achieving complete loss of Dicer [86]. Initially, Wang et al. [89] observed the effects of systematically knocking down *DICER1* expression in a number of pancreatic cancer cell lines (BxPC-1, Capan-2, Panc-1) on the expression of oncogenes and tumor suppressor genes. In all three cell lines, knocking out *Dicer1* caused variations in the expression profiles of the Kras oncogene as well as two tumor suppressors, including p53 and PTEN. A decrease in Dicer1 expression slowed the development of BxPC-1 tumors but increased that of Panc-1 tumors, with no discernible impact on Capan-1 neoplasms. In the future, it is probably essential to have tailored treatments for pancreatic cancer because distinct subtypes are caused by distinct molecular pathways [89]. We do not yet know whether these discoveries will lead to endocrine neoplastic processes of the pancreas. Even if there are not many pancreatic tumors that start in the endocrine system, we still need to undertake additional research to find out if DICER1 is a good therapeutic target.

### 9.3. Future Directions

Since its recent discovery, our understanding of DICER1 syndrome has expanded greatly; yet there is a pressing need for additional research into potential treatments and management strategies. Model experimental animal studies on Dicer have sought to identify effective ways of increasing levels of Dicer and related proteins to counteract the consequences of DICER1 syndrome, since this insufficiency is caused by *DICER1* gene mutations. Blandino and colleagues performed one such trial, in which they administered metformin to diabetic mice, which decreased the cancer incidence [90]. Mice given metformin had a decrease in tumor growth, an increase in DICER1 gene expression, and an elevation of microRNAs. A related and more recent study corroborated these findings; it demonstrated that metformin increases Dicer levels in both mice and humans by changing the location of AUF1, a protein that binds to *DICER1* mRNA and down-regulates DICER1, therefore increasing the stability of *DICER1* mRNA [91,92]. Metformin and similar compounds may help patients with a single functional *DICER1* allele, as it is possible to target and upregulate this single allele to achieve increased Dicer protein production. However, these treatments may not work for patients with biallelic *DICER1* gene mutations. A nonsense mutation in the DICER1 gene has been found in certain families; this mutation causes nonsense-mediated RNA degradation, shortened proteins, or stop codons in the coding sequence.

Antibiotics can restore transcription and translation of otherwise unintelligible sequences by encouraging the read-through of comparable premature stop codons, which causes tumors to develop in both cancerous and non-cancerous forms [93]. Human cells with known nonsense mutations were treated with the pharmaceutical medicine Ataluren, which is used to treat genetic problems, in one study to demonstrate this new technique. All three nonsense codons in the mutant alleles were read-through by Ataluren [94]. An updated study expanded on these results by changing the amount and placement of fluorines in Ataluren, which improved the read-through capability of stop codons in nonsense mutations [95]. Patients with DICER1 syndrome caused by nonsense mutations similar to those already studied may benefit from using Ataluren or comparable chemicals, while no research has been conducted on *DICER1* nonsense mutations using these medications. Although there has been a lot of focus on treatments for pleuropulmonary blastoma and cystic nephroma, further research is needed to understand the genetic abnormalities and acquired somatic mutations that cause these aberrant forms.

More questions may need to be answered [3]. In order to generate short RNAs of specific lengths, how do noncanonical Dicers, like Dictyostelium’s DrnA and DrnB, identify and cleave substrates? Do protists have any additional Dicer architectures? Dicer is the only gene in *C. elegans* that can encode both miRNAs and siRNAs. What is the difference or similarity in the recognition and cleavage of lengthy dsRNA and pre-miRNA substrates? In other words, does Dicer mainly interact with pre-miRNAs via the PAZ domain, or does it recognize and load long dsRNA into the enzyme through the helicase domain? Some of these questions could be answered by adding cryo-electron microscopy structures of *C. elegans* Dicer with both substrates to current study designs [3]. Moreover, as a result of binding Dicer, how does AGO choose 5p/3p miRNAs? In several model organisms, the exact mechanism by which AGO binds cleavage products and interacts with Dicer is yet unknown. Thermodynamic sensing encourages main strand selection and loading after Dicer releases a cleavage product. Nevertheless, further structural understanding of the RISC loading complex’s organization and the loading process is required [3]. Finally, is PACT (protein activator of Protein kinase R), the mammalian TARBP2 (Trans-Activation Responsive RNA-Binding Protein 2) paralog, physiologically significant, and to what extent are its activity and substrate specificity controlled by other Dicer-binding partners? These and other compelling questions remain on the table of biologists and geneticists and, hopefully, will probably be answered in this decade.

## 10. Conclusions

The range of diseases associated with DICER1 is extensive and we highlighted just some of them derived from our personal experience with iconographic support. The DICER1 human pathology remains incompletely understood. While specific diagnosis may require testing, no established histologic characteristics exist for more frequent lesions, such as MNG. From the viewpoint of an anatomical pathologist, pediatrician, and pediatric surgeon, a comprehensive assessment of the patient’s physical characteristics, such as chronological age and family history, will probably be the sole indicators of a *DICER1* alteration.

An excellent foundation for understanding future structures and/or their predictions can be found in structural investigations of Dicers with canonical Dicer topologies, which show traits linked to miRNA and siRNA formation. Additional distinct Dicer adaptations for miRNA synthesis are anticipated to be discovered in future investigations of Dicer in mollusks, nematodes, and other species. On the other hand, there are creatures that depend on Dicers, which just keep the core module working properly. How these Dicers produce tiny RNAs with a specific length is still a mystery. There may be more ways for Dicer to biogenesize short RNAs in the future, according to AlphaFoldbased models [3,96], which provide intriguing insights into the projected structural organization of these different Dicers. Grasping the intricate and crucial function of DICER1 in the human body can radically revolutionize our comprehension of the development of several illnesses. Further investigation into the related proteins of RISC, such as Drosha (another member of the RNase III superfamily) and the AGO family of proteins, as well as other molecules linked to RISC, has the potential to substantially uncover several goals for the treatment and surveillance of human diseases. Despite the obscure intricacy of these pathways in human cells, the widespread presence of DICER throughout eukaryotes suggests that animal research is still essential, in our opinion, and will serve as a mechanism to advance our bio-medical knowledge.

## Figures and Tables

**Figure 1 biology-14-00093-f001:**
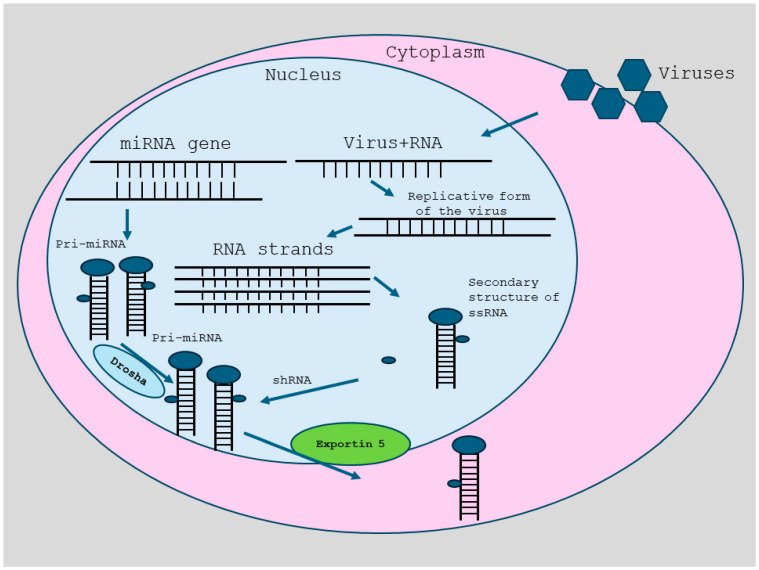
RNAi. RNA interference (RNAi), also known as PTGS or Post-Transcriptional Gene Silencing (PTGS) or Genetic Perturbation Platform, is a preserved biological mechanism that responds to dsRNA or double-stranded RNA, enabling resistance to both parasite of endogenous origin and exogenous harmful nucleic acids, while regulating the expression of genes, which codify proteins. This innate process for sequence-explicit gene silencing has the potential to transform experimental biology. It may have significant applications in genomics and functional genomics, as well as therapeutic intervention and other fields. Endogenous activators of the RNAi path including foreign DNA or dsRNA of viral origin, inconsistent transcripts from repeating genomic sequences (e.g., transposons), and pre-miRNA (miRNA, microRNA). In plants, RNAi underpins virus-induced gene silencing (VIGS), indicating a significant role in disease resistance. Investigations on *C. elegans* (*Caenorhabditis elegans*) has proposed a potential method for the control of endogenous genes by the RNAi machinery. In mammalian cells, large double-stranded RNAs (>30 nt) typically elicit an interferon response. A streamlined model for the RNAi process consists of two stages. Both stages involve a ribonuclease enzyme. The first step is for the RNase II enzymes Dicer and Drosha to transform the trigger RNA (which could be dsRNA or miRNA primary transcript) into short interference RNA (siRNA). The RNA-induced silencing complex (RISC) is an effector complex that incorporates siRNAs in the following step. As the RISC is assembled, the siRNA is unwound, allowing the single-stranded RNA to hybridize with the mRNA target. Argonaute, an RNase H enzyme, destroys the target mRNA, resulting in gene silencing (Slicer). The messenger RNA stays uncleaved if the siRNA/mRNA duplex displays mismatches. When translational inhibition occurs, genes are silenced. This illustration depicting RNA interference was adapted from the NCBI website: RNA Interference (RNAi) [4,5,6,7,8,9]. Functional studies of the mammalian genome can show how genetic changes cause changes in phenotype, and the Genetic Perturbation Platform (GPP), formerly known as the RNA interference (RNAi) Platform, supports these investigations.

**Figure 2 biology-14-00093-f002:**
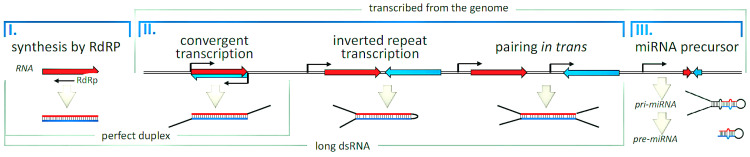
Various dsRNA. Schematic depiction of the characteristics and origins of various dsRNA substrates. There are two possible sources for genomic transcripts that produce double-stranded RNA (dsRNA): repeated transcripts in pathways that maintain genome integrity and genetic sequences in pathways that control gene expression (adapted from an Open Access source: Zapletal D, Kubicek K, Svoboda P, Stefl R. Dicer structure and function: conserved and evolving features. EMBO Rep. 2023 Jul 5;24(7):e57215. doi:10.15252/embr.202357215. Epub 2023 Jun 13. PMID: 37310138; PMCID: PMC10328071) [3].

**Figure 3 biology-14-00093-f003:**
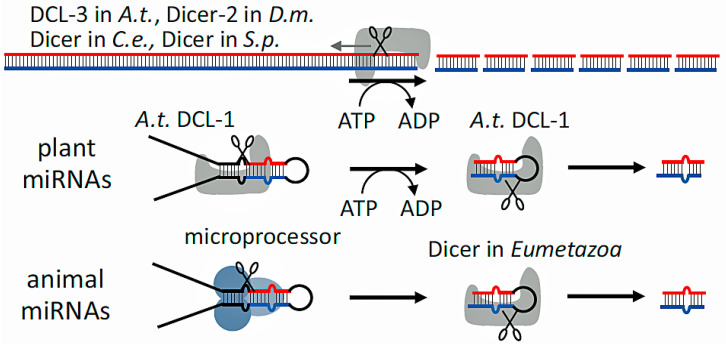
Substrate Cleaning. There are two ways to clean the substrate. During its processive mode, Dicer “feeds” its substrate by slicing lengthy dsRNA molecules in a series of sequential steps that are powered by the helicase domain’s ATP activity. Dicer attaches a new substrate after performing a single cleavage when in distributive mode (adapted from an Open Access source: Zapletal D, Kubicek K, Svoboda P, Stefl R. Dicer structure and function: conserved and evolving features. EMBO Rep. 2023 Jul 5;24(7):e57215. doi:10.15252/embr.202357215. Epub 2023 Jun 13. PMID: 37310138; PMCID: PMC10328071) [3].

**Figure 4 biology-14-00093-f004:**
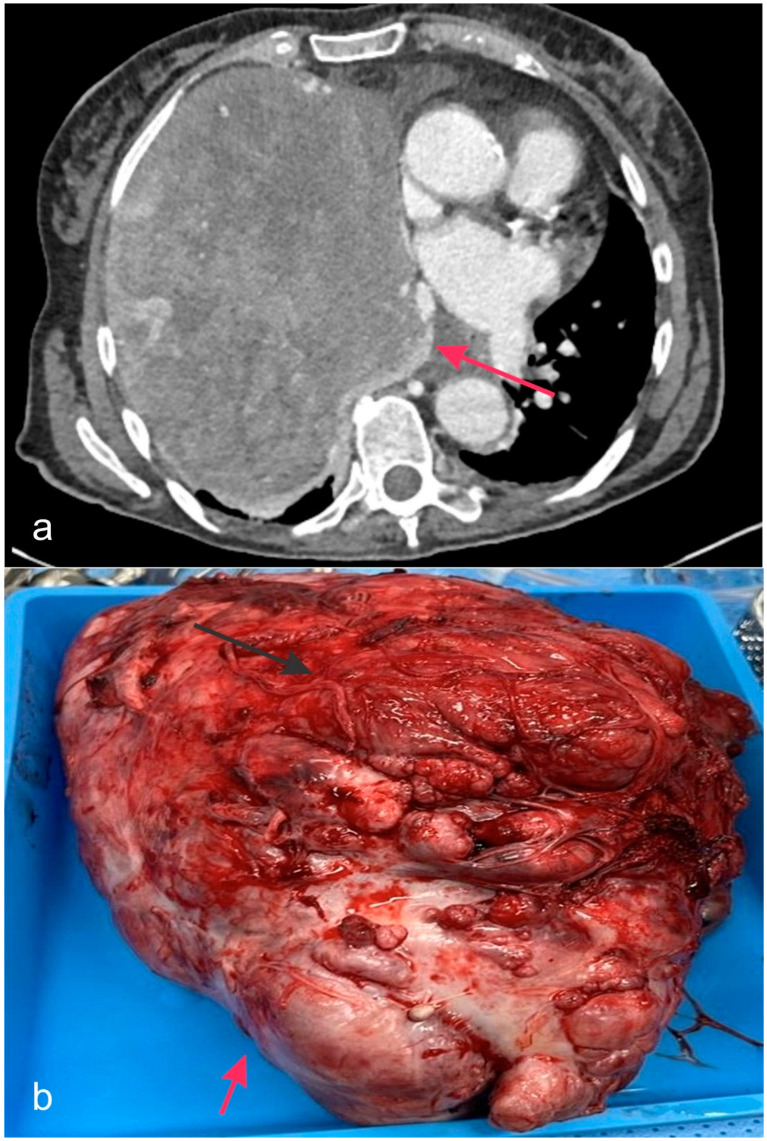
CT Imaging and Macroscopic View of Pleuropulmonary Blastoma. (**a**) CT scan of a 21-year-old female patient with a pleuropulomonary blastoma causing a mediastinal shift (arrow pointing to the mediastinal shift). (**b**) The figure shows the resected pleuropulmonary blastoma (arrow pointing to the bulk of the tumor) just after surgery. No copyright issue. The images come from the personal archive of Dr. F. Minervini.

**Figure 5 biology-14-00093-f005:**
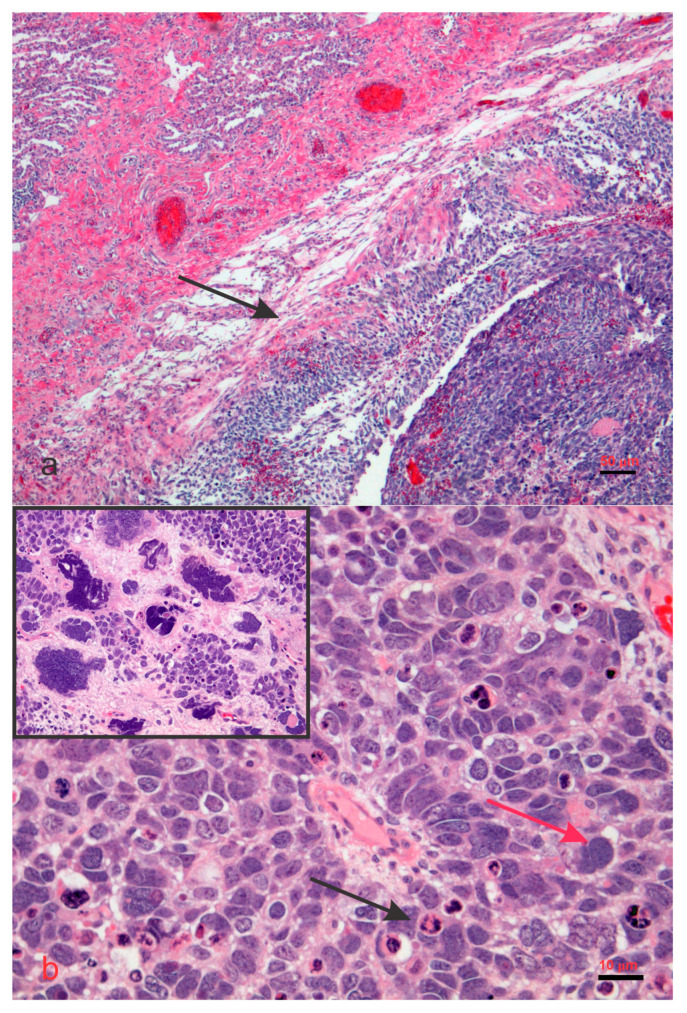
Microphotographs of Pleuropulmonary Blastoma. A heterogeneous solid tumor (**a**) abutting (arrow) the fibrotic pleura showing on higher magnification blastema-like area (arrow) infiltrating the soft tissue and exhibiting small-to-medium sized cells to very large cells (round, ovoid, slightly spindle) with hyperchromatic nuclei, high nucleus to cytoplasm ratio, and frequent mitotic bodies (**b**). In this case, foci of very large anaplastic cells (red arrow) with pleomorphic nuclei and mitotic figures as well as apoptotic figures (black arrow) were also observed (figure and inset). Hematoxylin and eosin staining, scale bar embedded in the microphotographs. No copyright issue. The images come from the personal archive of Dr. C. Sergi.

**Figure 6 biology-14-00093-f006:**
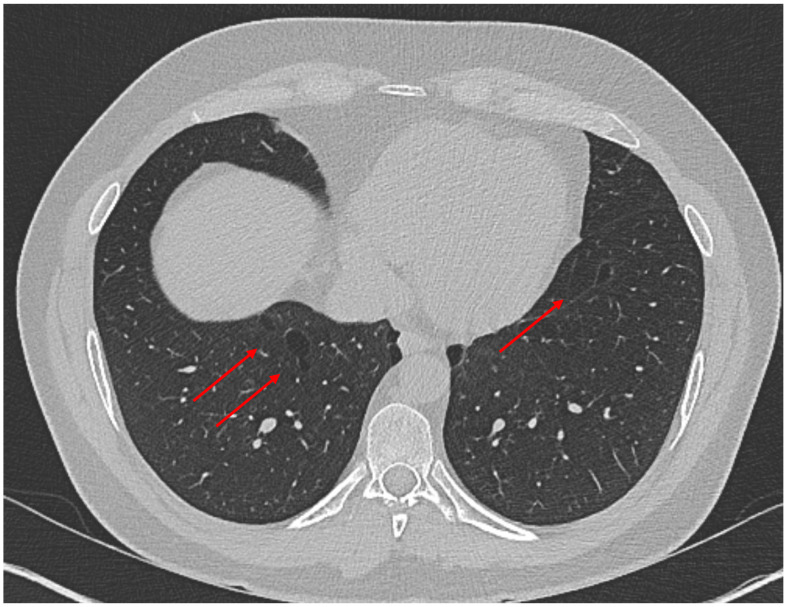
DICER1 associated lung cysts (arrows) in an asymptomatic 22-year-old male patient. No copyright issue. The image comes from the personal archive of Dr. F. Minervini.

**Table 1 biology-14-00093-t001:** Main DICER Roles in Pediatric and Youth Pathology.

Time	Main Mechanism(s)
Early Development of the Embryo	Cell cycle kinetics, Caspase 3 activation and apoptosis
Neural Progenitor-type Cells Proliferation	Cell cycle kinetics
Neuronal cell migration and differentiation	Cell cycle kinetics
Astrocyte differentiation	Cell cycle kinetics
Limb morphogenesis	Caspase 3 activation and apoptosis (programmed cell death))
Immunologic Competency	B lymphocytic cellular function, Stress granule response
Skin development	Cell cycle kinetics

**Table 2 biology-14-00093-t002:** Dysregulation or molecular alteration of DICER1 connected in Human Neoplastic Diseases [17].

Tumor	DICER1 Alteration (Red = Down-Regulation/Green = Up-Regulation)
NSCLC	
CRC	
Prostate Ca	
Breast Ca	
ALL	
Ovary Ca	
Ovary Ca	
Thyroid Ca	
Thyroid Ca	

Notes: Reduced expression is red, while increased expression is green. ALL—acute lymphoblastic leukemia; NSCLC—non-small cell lung carcinoma; CRC—colorectal carcinoma. In Trinucleotide repeat disorders, such as Huntington’s disease, Myotonic dystrophy type 1, and Fragile X syndrome, both normal and increased expressions have been recorded. Autoimmune disorders (e.g., ankylosing spondylitis, rheumatoid arthritis, multiple sclerosis, and autoimmune thyroid disease) are overwhelmingly reporting a reduced expression, apart from psoriasis. Other diseases, such as chronic stress, depression, Parkinson’s disease, and atherosclerosis, have shown reduced expression.

## Data Availability

All data are included in the manuscript.

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
